# Exploring the Role of Metabolites in Cancer and the Associated Nerve Crosstalk

**DOI:** 10.3390/nu14091722

**Published:** 2022-04-21

**Authors:** Inah Gu, Emory Gregory, Casey Atwood, Sun-Ok Lee, Young Hye Song

**Affiliations:** 1Department of Food Science, Division of Agriculture, University of Arkansas, Fayetteville, AR 72704, USA; inahgu@uark.edu (I.G.); cspierce@uark.edu (C.A.); 2Department of Biomedical Engineering, University of Arkansas, Fayetteville, AR 72701, USA; eagregor@uark.edu

**Keywords:** metabolites, lactate, amino acid metabolism, vitamins, cancer, cancer–nerve crosstalk, perineural invasion, tumor innervation

## Abstract

Since Otto Warburg’s first report on the increased uptake of glucose and lactate release by cancer cells, dysregulated metabolism has been acknowledged as a hallmark of cancer that promotes proliferation and metastasis. Over the last century, studies have shown that cancer metabolism is complex, and by-products of glucose and glutamine catabolism induce a cascade of both pro- and antitumorigenic processes. Some vitamins, which have traditionally been praised for preventing and inhibiting the proliferation of cancer cells, have also been proven to cause cancer progression in a dose-dependent manner. Importantly, recent findings have shown that the nervous system is a key player in tumor growth and metastasis via perineural invasion and tumor innervation. However, the link between cancer–nerve crosstalk and tumor metabolism remains unclear. Here, we discuss the roles of relatively underappreciated metabolites in cancer–nerve crosstalk, including lactate, vitamins, and amino acids, and propose the investigation of nutrients in cancer–nerve crosstalk based on their tumorigenicity and neuroregulatory capabilities. Continued research into the metabolic regulation of cancer–nerve crosstalk will provide a more comprehensive understanding of tumor mechanisms and may lead to the identification of potential targets for future cancer therapies.

## 1. Introduction

Cancer places a heavy burden on the global population as the second leading cause of death, with severe detriment in countries without access to sufficient medical care [1]. In 2020 alone, the World Health Organization’s GLOBOCAN reported that over 10 million men and 9 million women were diagnosed with some form of cancer [2]. The most prominent cancers among women include breast, colorectal, lung, cervical, and thyroid; while lung, prostate, colorectal, stomach, and liver cancers are the most common among men [2]. Though genetic factors may play a role in cancer onset, a heavy emphasis has been placed on the preventability of cancer, as highlighted by the increased probability of disease onset by smoking, sun exposure, and virus and bacterial incubation [3,4]. Over the last 20 years, increasing evidence has implicated diet as a major contributor in up to 20% of cancer cases [5]. This is made even more apparent by the variance in cancer dominance between countries and cultures [6] and the incidence of cancer among populations without access to proper nutrition [7].

The role of an altered metabolic state in which cells exhibit an increased conversion of glucose into lactate even in highly oxygenated areas, denoted the Warburg effect, has been an accepted hallmark of cancer since the 1920s [8,9]. This reliance on aerobic glycolysis leaves cells with a large deficiency in ATP and increased by-products such as reactive oxygen species (ROS) [9,10,11]. For cancer cells, maintaining ROS homeostasis via glutathione is essential, as moderate levels are proven to contribute to tumorigenesis and invasion while an overabundance leads to cell and DNA damage [10,12,13]. Moreover, increased ROS leads to upregulation of glutathione to maintain homeostasis; this increased defense can allow cancer cells to become resistant to therapies that rely on the induction of oxidative stress [14]. As the catabolism of glucose leaves them starved of energy, cancer cells can become dependent on glutamine for ATP [15,16,17,18]. Glutamine plays a vital role in macromolecule synthesis through the tricarboxylic acid (TCA) cycle (Figure 1) [16,19,20]. Therefore, glucose, glutamine, and their by-products are prevalent in studies searching for metabolic targets in cancer therapies [21,22,23,24]. 

Alternatively, researchers have come to acknowledge the role of lactate not only as a waste product but also as the driving force of mitochondrial respiration in oxidating cancer cells [25]. Thus, there has been continued interest in the investigation of metabolites in cancer progression. Amino-acid products of the TCA, urea, folate, and methionine cycles as well as those exogenously supplemented have proven important in facilitating cancer-cell protein and DNA synthesis as well as waste (e.g., ROS and nitric oxide synthase (NOS)), which progresses tumorigenesis [11,26]. Recent studies have targeted many amino acids to inhibit cancer progression, as many induce adaptive resistance to chemotherapy and radiation-therapy treatments [27,28,29,30,31,32]. In order to slow cancer progression, some clinicians have even suggested patients adhere to an “anti-cancer diet” to starve tumor cells of key nutrients [33,34]. However, many cancers alter their reliance on specific metabolites, rendering dietary restrictions ineffective, which has caused researchers to investigate pathways via targeted therapies [35,36]. Alternatively, some vitamins (e.g., vitamins A, C, and D) have been identified as antitumorigenic agents, presenting them as promising treatments in inhibiting tumor growth, while the upregulation of others (e.g., vitamin B_12_) is associated with decreased patient outcomes [37,38,39,40,41,42]. Continued examination of the contributions of lactate, amino acids, and vitamins in cancer mechanisms may provide more significant therapeutic options.

Within the past two decades, researchers have also acknowledged the interconnection of the nervous system with cancer progression and metastasis [43,44,45,46,47,48]. This cancer–nerve crosstalk occurs in cancers of highly innervated areas, including the pancreas [49,50,51,52], head and neck [53,54,55], colon and rectum [56,57,58], breasts [59,60,61,62], prostate [63,64,65], cervix [66,67,68], lungs [69], etc. There are two established subtypes of cancer–nerve crosstalk: perineural invasion (PNI) occurs when cancer cells migrate into the perineural layer of adjacent nerves [70,71,72] while its relative opposite, tumor innervation (TI), describes the phenomenon in which neurites extend into solid tumors [73,74]. PNI has been deemed a prognostic factor of select cancers, including cervical [68], liver [75], and colorectal [48], as well as a signifier of tumor recurrence in breast [61] and prostate cancers [76]. However, there still lack effective treatments of either form of the crosstalk, as the lack of sufficient pre-surgical screenings for PNI and TI complicate resection [77] and radiation and chemotherapeutic options are limited [78,79]. It is vital that there be continued exploration into the mechanisms of cancer–nerve crosstalk, as both PNI and TI contribute to progressive cancer-related pain, metastases, and worsened patient prognosis [74].

Previous studies have presented neurotransmitters and neurotrophins (e.g., substance P [80], nerve-growth factor (NGF) [81,82,83] and brain-derived neurotrophic factor (BDNF) [84]), chemokines (e.g., CX3CL1 [85] and CCL2 [86,87]), extracellular vesicles [66,88], and Schwann cells [52,89] as key players in cancer–nerve crosstalk. Moreover, glutamate, the product of glutaminolysis, acts as a potent central nervous system neurotransmitter that facilitates PNI in endometrial cancer [90] and TI in gliomas [57]. Glucose also works alongside neurotransmitters to modulate cancer metabolism and immune cell activity in the tumor microenvironment [91], while hyperglycemia in patients with type 2 diabetes mellitus correlates with an increased incidence of PNI in pancreatic cancer [92,93]. However, because there is presently a strong focus on targeting glutamine and glucose metabolism in cancer, this review will focus on the less-appreciated metabolites in cancer–nerve crosstalk that may be targeted to advance cancer therapeutics. As known contributors to cancer, nutrients should now be investigated in the context of cancer–nerve crosstalk. 

Using literature obtained via PubMed and Google Scholar databases (published in English through March 2022), this review will highlight select nutrients and their anti- and/or pro-cancer effects on metabolism, invasion, and metastasis. As discussed above, there exist many reviews that highlight the value of glutamine, glucose, and their by-products in cancer; therefore, this review will focus on the roles other key nutrients play in cancer metabolism and aggressiveness. Additionally, we identify potential target metabolites as key players in cancer–nerve crosstalk based on their roles in cancer, as well as their known mechanistic contributions to neurite outgrowth and nerve regeneration. Further investigation into these nutrients will provide vital insights to develop targeted therapies and promote improved global patient outcomes.

## 2. Metabolites and Cancer

### 2.1. Background

In 1924, Otto Warburg published his findings, introducing dysregulated metabolism as a hallmark of cancer [94]. This altered metabolic state, aptly titled the Warburg effect, is described by the phenomenon in which cancer cells increase their uptake of glucose and production of lactate with a heavy reliance on aerobic glycolysis [94]. In this metabolic state, cancer cells produce very limited ATP via glycolysis and may become dependent on glutamine as an energy source [21]. Therefore, suppression of glucose and glutamine metabolism is heavily investigated in cancer therapeutics [21,22,23,24]. 

Lactate has gained a new appreciation for its contribution to oxidative respiration under normoxic conditions [95]. The discovery that lactate is both a waste product and vital tumor energy source highlights that metabolites other than glucose and glutamine may promote an environment fit for cancer-cell growth and proliferation. Downstream amino-acid by-products, e.g., asparagine [96], arginine [97], cysteine [98], serine [99], and glycine [100], have also been investigated for their contribution to the survival of cancer cells. While the deprivation of these nutrients appears to be an effective treatment in some cases [101,102], more investigation into therapies targeting each metabolic pathway is needed. Alternatively, some amino acids and key vitamins, e.g., vitamins A [37], B [103], D [104], E [105], K [106], and C [107], act as antitumorigenic factors and suppress cancer progression. To bring attention to these relatively underappreciated metabolites, this review highlights lactate, vitamins, and amino acids for their roles in progressing, suppressing, and preventing cancer (Table 1).

### 2.2. Lactate

Lactate is the major metabolite of glycolysis, which is catabolized from pyruvate by lactate dehydrogenase (LDH) (Figure 1) [94,137]. The primary isomer of lactate produced in humans is L-lactate, while D-lactate is synthesized at 1 to 5% the concentration of L-lactate [131,138]. Therefore, this review will focus on L-lactate unless stated otherwise. In healthy tissues, endogenous lactate levels are around 1.5 to 3 mM; however, under Warburg effect-like conditions, cancer cells contain 30 to 40 mM lactate [131]. In the past, lactate was recorded only as the waste product of glycolysis under hypoxic conditions; however, within the last 2 decades, researchers have gained an appreciation for its facilitation of oxidative metabolism in oxygenating cancer cells [25]. As such, when fed glucose-rich media with 10 mM sodium lactate supplementation, cervical cancer cells favored lactate and switched from glycolysis to lactate oxidation [25]. These cells continued to thrive following glucose starvation [25]. Conversely, glycolytic colorectal cancer cells had an adverse reaction when glucose was removed from their media [25]. In the oxidative cervical cancer cells, mRNA for monocarboxylate transporter-1 (MCT1), a protein that imports lactate, is transcribed at a higher rate than MCT4 mRNA, the protein exporter of lactate. MCT4 exhibited higher expression versus MCT1 in glycolytic colorectal cancer cells [25]. The study concluded that MCT1 and hypoxia are mutually exclusive in glycolytic tumors, and MCT1 inhibition induced lung carcinoma cell necrosis and sensitizes cells to irradiation in vivo [25]. One breast cancer study found that endogenous lactate receptor G protein-coupled receptor 81 (GPR81) mRNA expression is elevated in clinical estrogen receptor alpha-positive (ERα^+^) and human epidermal receptor-growth factor 2-positive (Her2^+^) breast cancer versus triple-negative breast cancer (TNBC) and benign tissue samples [139]. In vitro, GPR81 regulates MCT1 expression and lactate uptake in MCF-7 breast epithelial breast cancer cells and is responsible for cancer-cell proliferation and survival when lactate is the main nutrient source [139].

Bonuccelli et al. found that 10 mM L-lactate acted as a chemoattractant to almost double breast cancer-cell migration versus untreated cells [129]. In vivo, mice treated with L-lactate at 500 mg/kg of body weight (BW) increased lung metastases by 10.6-fold versus control mice [129]. Apicella et al. cultured non-small-cell lung cancer (NSCLC) tumors ex vivo before being injected into mice resistant to mesenchymal-to-epithelial transition (MET) tyrosine kinase inhibitors (TKI) [132]. This study found that resistant tumors upregulated lactate production and nuclear factor kappa-B (NF-κB), which induced hepatocyte growth factor (HGF) transcription by adjacent cancer-associated fibroblasts [133]. This pathway was shown to contribute to tumor resistance to MET TKI; stromal HGF and tumor MCT4 were also upregulated in epidermal growth-factor receptor TKI-resistant tumors of NSCLC patients [132].

In a genetic evaluation of in vitro breast cancer, MCF-7 cells were incubated with 0 to 20 mM exogenous lactate and analyzed for the expression of proto-oncogenes (i.e., NRAS, MYC, and phosphatidylinositol-4,5-bisphosphate 3-kinase catalytic subunit alpha PIK3CA), proliferative genes (i.e., ATM, CCND1, CDK4, CDK1A, CDK2b, AKT1, and MIF), tumor suppressors (i.e., BRCA1 and BRCA2), and transcription factors (i.e., HIF1A and E2F1) [130]. Following 48 h, cells treated with 4500 mg/L glucose (endogenous lactate) upregulated oncogenes, proliferative gene expression (except for MIF), tumor-suppressor genes, and transcription factors by over 2-fold [130]. Oncogenes, transcription factors, tumor suppressors, and proliferation genes were upregulated after 6 h of 10 mM treatment [130]. Expression was lower, yet still significant, after 48 h for all genes except for BRCA1 [130]. Following 6 h of 20 mM exogenous lactate therapy, MYC, transcription factors, tumor suppressors, and proliferation genes except ATM and CDK2b increased in expression [130]. Similarly, all but NRAS, PIK3CA, ATM, and CDK2b slightly decreased between 6 and 48 h [130]. The authors concluded that lactate changes cancer-cell transcription and acts as both a pro- and antitumorigenic factor in in vitro breast cancer [130].

A variety of therapies have been investigated in targeting glycolysis and lactate. Docetaxel-resistant prostate cancer cells were sensitized to chemotherapy following treatment with sodium oxamate, a structural analog to pyruvate which inhibits LDH and glycolysis [30]. The study found success in targeting LDH-A expression in vitro [30]. In papillary thyroid carcinoma patients, LDH-A mRNA and protein levels correlate with an aggressive phenotype and poor prognosis [140], while also associated with high glycolytic activity, radioresistance, and poor survival of NSCLC patients [133]. LDH-A promotes migration, proliferation, and epithelial-to-mesenchymal transition (EMT), regulates tumorigenicity and autophagy via the AMP-activated protein kinase (AMPK) pathway, and inhibits apoptosis of thyroid cancer in vitro and in vivo [140]. LDH-A inhibition sensitizes NSCLC cells to radiation therapy by blocking cellular energy metabolism and increasing X-ray-induced DNA injury via ROS production [133]. Inhibition, alone [140] and in conjunction with irradiation [134], induces apoptosis and autophagy of cancer cells.

Alternatively, D-lactate dimers (DLAD) have been found to be cytotoxic in some neuroblastoma and cervical cancer studies; therefore, Dikshit et al. utilized DLAD 3-(4,5- dimethylthiazol-2-yl)-2,5 diphenyl tetrazolium bromide to evaluate melanoma cell growth in dose-dependent manner (1.25 to 10 mg/mL) [131]. Following only 1 day of 10 mg/mL treatment, melanoma cells experienced complete cytotoxicity [131]. DLAD was found to contribute to dose-dependent inhibition of mitochondrial function, with inhibitory effects on cell growth even with pH neutralization [131]. In vivo, melanoma xenografts regressed in size and showed significantly decreased proliferation over a 21-day period. Furthermore, immunohistochemistry for innate immune cells indicated a significant increase in immune response and tumor growth inhibition [131]. With minimal metabolic effect on fibroblasts versus squamous-cell carcinoma cells in vitro, there exists promise for DLAD treatment in targeting tumor lactate metabolism [131].

### 2.3. Vitamin A

Vitamin A is a group of fat-soluble retinols and their derivatives, including retinyl ester, retinal, retinoic acid, and pro-vitamin A carotenoids, including α- and β-carotene [108]. Retinol is mostly acquired from the diet in precursor forms, e.g., retinyl esters from animal sources and pro-vitamin A carotenoids from plants [141]. 

Since retinoids have an important role in regulating cell growth, proliferation, and differentiation, studies have investigated the correlation between vitamin A and cancer patient prognosis [142,143]. A meta-analysis with 31 studies highlighted an inverse association between vitamin A intake and gastric cancer risk [108]. Another meta-analysis of 17 studies reported that higher blood concentrations of α- and β-carotene, total carotenoids, and retinol are significantly associated with decreased lung cancer risk [109]. Fulan et al. found an inverse association between the total intake of vitamin A/retinol and breast cancer risk [105]. Both natural and synthetic retinoids have been shown to exert chemotherapeutic effects in cancer with antiproliferative, proapoptotic and antioxidant activities [37]. Synthetic retinoid derivative fenretinide (4-HPR) has shown succuss in tumor-cell cytotoxicity in many cancers such as breast, prostate, colon, head and neck, and lung [37].

### 2.4. Vitamin B

Cobalamin (vitamin B_12_) is a water-soluble vitamin that is abundant in foods of animal origin [144]. Vitamin B_12_ can also be supplemented in foods in which it is not naturally synthesized. For humans to absorb vitamin B_12_ from food, hydrochloric acid in the stomach must separate B_12_ from the protein to which it is attached and then should combine with intrinsic factors to be absorbed by the body [145]. Analogs of cobalamin include: methylcobalamin (MeCbl), adenosylcobalamin (AdCbl), hydroxycobalamin (OHCbl), and cyanocobalamin (CNCbl). MeCbl and 5-deoxyadenosylcobalamin are active in human metabolism, while OHCbl is found in foods and is used as the first line of treatment for cyanide poisoning. CNCbl is a synthetic B_12_ analog used in food fortification and supplementation [146]. Vitamin B_12_ plays an important role in two biochemical reactions in humans: the formation of methionine by methylation of homocysteine (Figure 1), by which tetrahydrofolate, a precursor of purine and pyrimidine synthesis, is produced [147] as well as the conversion of L-methylmalonyl coenzyme A into succinyl coenzyme A, which is essential for the formation of the neuronal myelin sheath [148].

There has been some evidence of a relationship between intake of vitamin B and cancer [149]. In a review of the United Kingdom’s Health Improvement Network primary-care database, elevated plasma B_12_ levels were linked with a higher 1-year cancer risk, particularly for myeloid, liver, and pancreatic cancers. This suggests that some cancers may affect B_12_ metabolism [113]. Cancer caused by chromosomal breaks from the incorporation of uracil instead of the appropriate base is linked to folate deficiency and potentially vitamin B_12_ and B_6_ deficiencies [150]. In a nested study, higher blood B_12_ levels were associated with increased overall lung cancer risk [114]. One meta-analysis, however, showed that vitamins B_1_, B_3_, B_6_, and B_9_ are associated with reduced risk of esophageal cancer, while vitamin B_12_ was associated with induced esophageal cancer risk [103].

Furthermore, the decrease in thiamine (vitamin B_1_) transporter gene SLC9A3 has been evident in in vitro breast cancer cells as well as breast and gastric tumors [111,112,151]. Ng et al. highlighted a significant increase in plasma SLC9A3 methylation levels in cancer patients versus healthy participants, resulting in thiamine deficiency in patients [112]. In an in vivo study of breast cancer, mice were supplemented with 0 to 2500 times the recommended daily amount to combat this deficiency; mice given 12.5 to 75 times the recommended dose exhibited a significant increase in cancer-cell proliferation versus untreated mice [110]. However, it was also noted that oversupplementation, i.e., mice given 2500 times the recommended dosage, saw a decrease in tumor growth, suggesting the potential of thiamine overdose as a cancer therapy [110]. 

Thiamine is a coenzyme of pyruvate dehydrogenase (PDH), which converts pyruvate into acetyl-co-A to connect glycolysis and the TCA cycle [152,153]. However, similar to thiamine, the PDH subunit PDH-E1β is downregulated in breast cancer and HeLa cervical cancer cells as well as in vitro mouse fibroblasts under prolonged hypoxia [154]. This reduction in PDH was found to induce aerobic glycolysis and lactate production [154]. Therefore, the authors concluded that the activation status of PDH-E1β informs cancers cells whether to perform in a Warburg effect-like metabolic state (when silenced) or to enhance tumor growth (when upregulated) [154]. Interestingly, the group also found that bladder, melanoma, ovarian, and prostate cancer patients presented with high ratios of PDH-E1β, with prostate cancer patients experiencing up to a 50% amplification in total PDH expression. 

### 2.5. Vitamin C

Ascorbic acid (vitamin C) is a water-soluble compound rich in fruits and vegetables. Unlike other species, humans need to consume vitamin C from external sources, as humans do not have gulonolactone oxidase to synthesize endogenous vitamin C [155]. Vitamin C is an important antioxidant and reducing cofactor in many enzymatic reactions related to collagen synthesis, tyrosine metabolism, hypoxia-inducible transcription factor (HIF) regulation, and epigenetic regulation of histone and DNA demethylation [156].

Due to its antioxidant activities, vitamin C has been widely researched as an anticancer agent for decades [38]. However, the use of vitamin C is cancer therapeutics is limited, as its anticancer effects are only observed in vitro [38,39]. However, it was recently shown that a high dose of intravenous vitamin C selectively reduces the proliferation and growth of cancer cells, with limited interference in healthy cell metabolism [108,157,158]. Yang et al. showed that high concentrations (1 to 5 mM) of vitamin C significantly inhibit cell proliferation and increase apoptosis in melanoma cells, while low concentrations (100 µM) support cell growth and migration [116]. Daily oral administration of low concentrations of vitamin C (0.0075 g/g of BW) promotes significant melanoma tumor growth in mice (*p* < 0.05) compared to control mice [116]. Zeng et al. treated breast cancer cells with low, intermediate, and high doses of vitamin C (0.01 to 2 mM) and demonstrated that low and intermediate doses induce cell migration and invasion while high doses suppress EMT [115]. High doses of vitamin C (4 g/kg of BW) also significantly suppressed the metastasis of breast cancer in a xenograft mouse model [115]. These studies suggest that pharmacological doses of vitamin C over 0.1 mM may increase the production of ROS and DNA damage in cancer cells, while physiological doses promote the tumor growth and metabolism [158]. Although potential antitumorigenic mechanisms of high-dose vitamin C (e.g., hydrogen peroxide generation, enzymatic cofactor reactions, antioxidant, and anti-inflammatory activities) have been proposed, more clinical investigations are needed to support the potential of vitamin C as an effective cancer-therapy option [159,160,161,162].

### 2.6. Vitamin D

Vitamin D is a precursor to the fat-soluble steroid hormone calcitriol (1, 25(OH)_2_D_3_) [163]. Vitamin D_3_ (cholecalciferol)and vitamin D_2_ (i.e., ergocalciferol) can be synthesized by exposing skin to ultraviolet light or consumed from fortified food and supplements [104]. Vitamin D is hydroxylated into 25-hydroxyvitamin D (25(OH)D) in liver and is activated to produce calcitriol in the kidneys [164]. Calcitriol notably regulates calcium-phosphate homeostasis for bone health [165]. Upon binding to the vitamin D receptor (VDR), calcitriol regulates gene transcription related to cellular growth, differentiation, apoptosis, and the function of the immune, nerve, and cardiovascular systems [165,166].

Decades of research have demonstrated that low serum 25(OH)D is related to the initiation of many cancers, such as colon, breast, prostate, and gastric [164,167]. Accumulating evidence indicates that calcitriol has antitumorigenic effects through its antiproliferative, proapoptotic, antiangiogenic, and antimetastatic activities in various cancers [104,164,168]. Calcitriol and vitamin D_3_ analogs suppress the secretion of proteolytic enzymes in breast and prostate cancer cells while also inhibiting the expression of receptors of cell-adhesion molecules, e.g., vascular cell-adhesion protein 1 (VCAM-1) [117]. This is a noteworthy observation, as proteolytic enzyme activation and cell-adhesion-molecule expression in cancer cells can increase invasiveness and metastasis [117]. Some studies have shown that 1,25(OH)_2_D_3_ and vitamin D analogs have potential anticancer effects [40]. Chiang et al. compared the effects of 1,25(OH)_2_D_3_ and 19-nor-2-(3-hydroxypropyl)-1α,25-dihydroxyvitamin D_3_ (MART-10) on hypopharyngeal and tongue cancer cells [120]. MART-10 showed the greater inhibiting effect on both cancer-cell growth via G_0_/G_1_ cell-cycle arrest and the upregulation of p21 and p27 expression. Milczarek et al. reported that vitamin D analogs sensitize colon cancer to 5-fluorouracil in vivo [118]. Gorham et al. suggested a daily intake of 1000 to 2000 IU/day of vitamin D_3_ to decrease the risk of colorectal cancer [119]. 

### 2.7. Vitamin E

Vitamin E is a group of hydrophobic compounds in eight isoforms consisting of α, β, γ, and δ subtypes of tocopherols and tocotrienols [169]. Tocopherols are saturated forms of vitamin E, while tocotrienols are unsaturated with short tails and three double bonds [121]. They are rich in oils such as vegetable, palm, and rice bran, along with cereal grains [170]. After being absorbed in the small intestine, vitamin E, specifically α-tocopherol, binds to an α-tocopherol transfer protein in the liver [171].

Due to their lower bioavailability versus α-tocopherol, tocotrienols have been neglected for years [172]. However, it has been reported that some tocopheroland tocotrienol isoforms (i.e., γ and δ) have greater anticancer activities than the most abundant isoform, α-tocopherols [173]. In addition, there is increasing evidence that supports the anticancer effects of tocotrienols in breast, colorectal, lung, prostate, pancreas, and liver cancers [41]. γ-tocotrienol, for instance, increases cell-cycle arrest in G_1_/S phase in MCF-7 and MDA-MB-231 breast cancer cells [121,122] and induces apoptosis in HT-29 colon cancer cells by increasing apoptotic caspase-3 and downregulating NF-κB signaling [123]. Moreover, tocotrienols significantly sensitize cancer cells to chemotherapeutic drugs [174].

### 2.8. Vitamin K

Vitamin K is a group of lipid-soluble vitamins that has two natural forms: vitamin K_1_ (phylloquinone) and vitamin K_2_ (menaquinone) [175]. Vitamin K_1_ can be found in green leafy vegetables and olive and soybean oils, while vitamin K_2_ is mainly present in fermented food such as cheese, natto, and curds, endogenously generated by intestinal bacteria [176].

In the prospective cohort study of the European Prospective Investigation into Cancer and Nutrition-Heidelberg, dietary intake of vitamin K_2_ had an inverse association with cancer mortality [177]. An alternate prospective cohort analysis in a Mediterranean population also indicated that dietary intake of vitamin K is inversely associated with the risk of cancer mortality [175]. Vitamin K_2_ also increases nonapoptotic cell death with autophagy in TNBC cells [42]. However, the relationship between vitamin K and cancer is still unclear, and further studies are required to properly understand the mechanistic contributions of vitamin K to cancer.

### 2.9. Asparagine

Asparagine is a nonessential amino acid that is either synthesized from glutamate and aspartate by asparagine synthetase (Figure 1) or acquired from exogenous sources, e.g., cereal grains [178]. In recent years, asparagine, a precursor to the Class 2A carcinogen acrylamide, has been found to be important in maintaining the health of glutamine-independent liposarcoma and breast cancer cells [19,96,178]. As an exchange factor, asparagine works to import exogenous amino acids, e.g., arginine and serine, and 2 mM asparagine activates the mammalian target of rapamycin-1 (mTOR1) protein complex, a major complex that regulates cell growth and proliferation, in the absence of glutamine in HeLa cervical cancer cells [96,179]. Krall et al. noted that these functions promote asparagine-induced protein and nucleotide synthesis, contributing to the longevity of cancer cells [96]. Furthermore, Knott et al. demonstrated that while dietary asparagine restriction does not affect tumor growth, high-asparagine diets (4%) promote EMT and tumor metastasis in vivo versus low-asparagine diets (0.6%) [180].

Previously, studies focused more on the importance of L-asparaginase, a bacterial enzyme utilized as the first treatment of acute lymphoblastic leukemia and lymphoma [181,182,183,184]. One in vitro study found that silencing asparagine synthase increased ovarian cancer cells’ sensitivity to L-asparaginase up to 500-fold, concluding that asparagine synthetase may be used as a predictor of L-asparaginase therapy efficacy [181]. Unfortunately, the effectiveness of the treatment does not translate to in vivo models [185]. A more recent preclinical study showed that the administration of 20,000 U/kg asparaginase alone for 14 days insufficiently produced leukemia cytotoxicity and required additional glutaminase activity for both asparagine synthetase-positive and -negative mice [186]. Moreover, a metabolic analysis of 19 leukemia cells and 26 leukemia patients proved that cancer cells of lower mitochondrial respiration and glycolytic function are more sensitive to asparaginase therapy than higher glycolytic cells [187]. Continued investigation into targeting asparagine may contribute to more effective and clinically translatable cancer therapies.

### 2.10. Arginine

Arginine (2-amino-5-guanidinovaleric acid) is the semi-essential amino-acid precursor of nitric oxides, polyamines, and glutamate, which is upregulated in times of stress and directly activates mTOR [97,188]. In the urea cycle, citrulline and aspartate synthesize arginosuccinate by arginosuccinate synthase, arginosuccinate synthase synthesizes arginine by arginosuccinate lysate, and arginine synthesizes nitric oxide and citrulline by nitric oxide synthase (Figure 1) [97,189]. Recent studies have suggested that utilizing the presence of diminished plasma arginine levels acts as a biomarker of clinical prostate (<67.18 µmol/L) and breast cancers [126,190]. While not proposing a cutoff for diagnosis, Hu et al. reported that luminal A, luminal B, HER2^+^, and TNBC patients showed significantly lower arginine levels (7.34 ± 5.64, 9.98 ± 6.84, 8.27 ± 6.78, and 4.18 ± 3.34 µmol/L, respectively) compared to healthy patients [97,189,190]. Additionally, a diet rich in soy protein, fish, walnuts, and peanuts can supplement intracellular arginine synthesis [191], on which melanoma and ovarian cancer cells have been found to be dependent on in cases of arginosuccinate synthase deficiency [25,125]. Therefore, arginine starvation, which functions by inducing asparagine synthetase and depleting aspartate, has been used to treat arginosuccinate synthase-1-deficient breast cancer in vitro [124] and in vivo [101]. Ji et al. utilized PEGylated arginine deiminase 20,000 molecular weight (ADI-PEG20) to degrade arginine in ovarian cancer cells and xenograft models [125], while Izzo et al. found the therapy unsuccessful in a limited clinical study of liver cancer [192]. One in vitro study of ADI-PEG20-resistant melanoma reported that the oncogene c-MYC binds to the arginosuccinate synthase-1 promoter to upregulate arginosuccinate lysate and PI3K/AKT sensitivity [26]. mTOR signaling, glutamine, glutamine dehydrogenase, and sensitivity to glutamine inhibitors are also enhanced [26]. 

### 2.11. Serine and Glycine

Serine is a nonessential amino acid synthesized when 3-phospho-glycerate is oxidized to 3-phospho-hydroxypyruvate by phosphoglycerate dehydrogenase (PHGDH), transaminated to phosphoserine by phosphoserine aminotransferase (PSAT), and finally dephosphorylated via phosphoserine phosphatase (PSPH) [128,193,194]. Glycine, which is transformed from serine de novo in mitochondria via serine hydroxymethyltransferase (SHMT) and vitamin B_6_ (Figure 1) [128,136], is a nonessential amino acid and has been shown to work alongside serine in protein synthesis in 60 tumor-derived cell lines (NCI60) [195]. Dolfi et al. also highlighted that glycine exchange rates significantly correlate with cell proliferation and DNA synthesis in these 60 cancer-cell lines [195]. Both serine and glycine are highly regarded for their roles in protein, phospholipid, and glutathione synthesis via the serine synthesis pathway (SSP) [99,196,197]. As key players in one-carbon metabolism (i.e., folate and methionine metabolism), both serine and glycine collaborate in nucleotide synthesis, e.g., purine, in healthy patients [198,199]. However, while serine is accepted to contribute to cancer-cell purine synthesis, glycine has been demonstrated to work in a cancer-type-dependent manner [128]. For example, Labuschagne et al. demonstrated that in vitro breast and colon cancer cells prefer serine over glycine, and when starved of serine, cells show reduced nucleotide synthesis; furthermore, cells fed 0.4 to 2 mM serine exhibit increased proliferation [99]. The study found that cancer cells react to glycine in a dose-dependent manner: while low concentrations (0.4 mM) moderately increase cell proliferation, higher concentrations (1 to 2 mM) inhibit cancer cell proliferation [99]. Another study found that brain metastatic breast cancer cells upregulate de novo serine when deprived of the exogenous alternative, and PHGDH suppression reduces brain metastases of NCSLC and TNBC in vivo [134]. A recent in vitro genetic analysis found that when fed serine-free media, serine-starvation-resistant colon cancer cells increased serine, glycine, and threonine metabolic pathways [135]. In the short term, this serine inhibition decreases Yes-associated protein activation, which controls tumorigenesis, but does the opposite over a prolonged period, as SSP promotes the catabolism of PHGDH, PSAT1, and PSPH [135]. Furthermore, Meiser et al. found that serine–glycine catabolism is induced by stress and contributes to formate levels in in vivo colorectal cancer [136]. Interestingly, a study of NSCLC found that most de novo serine and glycine are allocated for glutathione synthesis [128], an important antioxidant in ROS homeostasis. Maddocks et al. proposed that serine and glycine inhibition must be used in parallel in vivo to induce a significant decrease in tumor size in colon cancer [102]. The study concluded that serine starvation in vitro induces cell metabolic stress and causes cells to rely on p53-mediated glycolysis [102].

As previously noted, SSP and serine–glycine metabolism enzymes are also implicated in cancer [193,197]. One breast cancer study found that PHGDH and PSPH are highly expressed in in vitro TNBC, and these, along with SHMT-1, are also elevated in stromal TNBC tumors [193]. Tumor PSPH positivity, stromal PSPH positivity, and stromal SHMT-1 negativity are linked to decreased survival in TNBC and HER-2 breast cancers [193]. Additionally, glycine decarboxylase, which transforms glycine into the one-carbon metabolism intermediate methylenetetrahydrofolate, was elevated in MDA-MD-453 and MDA-MB-435 breast cancer cell lines [193].

### 2.12. Cysteine 

Cysteine is the rate-limiting substrate in glutathione production in cells [200]. Cancer cells can become dependent on cysteine in order to uphold the functions of glutathione, e.g., ROS depletion, protein modification, and cell signaling [14,98]. Cysteine is regulated via the cystine–cysteine cycle, in which extracellular cystine is captured, imported via the cystine/glutamate antiporter system, and transformed into cysteine by thioredoxin reductase-1 (Figure 1) [14,201]. Because cancer cells produce an abundance of ROS as a product of aerobic glycolysis, increased levels of glutathione, and thus increased cysteine levels, are vital in maintaining ROS homeostasis to promote tumorigenesis [10,14]. While a recent study of in vitro and in vivo ovarian cancer found cysteine depletion to be a successful therapy [202], many patients do not respond to this method of induced oxidative stress [14,98]. Consequently, studies have investigated targeting the cystine/glutamate antiporter system [29,98]. Tarragó-Celada et al. evaluated liver-metastatic colon cancer in vitro and discovered that metastatic cell proliferation significantly decreases following cystine starvation, and cells are particularly sensitive to cystine/glutamate antiporter-targeted therapies [29]. Alothaim et al. demonstrated that inhibitors of histone deacetylase-6, a moderator of tumor-cell proliferation and metastasis, sensitize cystine/glutamate antiporter-targeted therapy-resistant TNBC cells to cysteine deprivations by signaling necroptosis and ferroptosis cell death [98]. Alternatively, one in vitro study utilized autophagy inhibition to diminish cysteine homeostasis via the deletion of the SLC7A11 cystine transporter gene in pancreatic cancer cells [127]. However, more clinically translatable studies are needed to develop the complete mechanistic understanding of cysteine in cancer and cysteine-targeted therapies.

## 3. Metabolites and Cancer–Nerve Crosstalk

### 3.1. Background

In the last two decades, an emergence of evidence has supported the role of the nervous system as a key player in cancer progression, increasing patient pain and poor outcomes [70,203]. PNI occurs when cancer cells invade adjacent nerves to aid in metastasis to secondary sites, with a high prevalence in cancer such as colorectal, head and neck, liver, pancreatic, and prostate [75,204]. TI, however, is the event in which neurites from adjacent nerves infiltrate nearby solid tumors, e.g., breast, cervical, head and neck, lung, and pancreatic cancers [62,66,69,205]. This crosstalk complicates cancer treatment, as there is limited success in preoperative diagnoses of nerve involvement in cancer, making surgical interventions more difficult to successfully complete [77], and few studies have explored radiation and chemotherapy as a method of blocking cancer and/or neurite invasion [78,79]. Though more studies have investigated the mechanism of PNI, it is evident that both forms of cancer–nerve crosstalk are progressed via chemokines, neurotrophins, and neurotransmitters [45]. However, the role of tumor metabolic dysregulation in cancer–nerve crosstalk remains underappreciated. Only within the last 5 years has literature immerged presenting metabolic players, e.g., vitamin C uptake gene SLC2A3 [206], asparagine synthetase [28], neuron-secreted serine [207], and lactate importer MCT1 [205], as contributors to PNI and/or TI. Alternatively, there is abundant literature examining the neuroregulatory properties of these metabolites. Combining this knowledge with the present understanding of tumor metabolism may aid in developing a more comprehensive understanding of cancer–nerve crosstalk mechanisms and in educating future potential therapies.

### 3.2. Known Contributors of Cancer–Nerve Crosstalk

#### 3.2.1. Vitamin C and SLC2A3

Vitamin C has an important role in synthesizing neurotransmitters. Recently, some studies showed the effect of vitamin C on peripheral nerve regeneration after traumatic injury [208,209]. In addition, Gao et al. reported that the solute carrier family 2 member 3 (SLC2A3) expression was remarkably associated with PNI in colorectal cancer (Figure 2A) [205]. SLC2A3 gene upregulation, which encodes glucose transporter 3, showed decreased disease-free survival in colorectal cancer patients [210]. Liu et al. showed that low SLC2A3 expression in acute myeloid leukemia significantly suppressed the effect of vitamin C, resulting in diminished overall survival [211]. Further investigation on tumor SLC2A3 expression should be conducted to develop a more comprehensive understanding of SLC2A3 and vitamin C in cancer–nerve crosstalk.

#### 3.2.2. Asparagine and Asparagine Synthetase

While the present literature does not highlight the importance of asparagine in the nervous system or neural disorders, the mutation of asparagine synthetase-promoting genes is linked to severe impairments. Asparagine synthetase deficiency is a congenital disorder that causes cognitive impairment, microcephaly, seizures, and progressive cerebral atrophy [212]. In a clinical and in vitro study of oral squamous-cell carcinoma, Fu et al. presented a correlation between high asparagine synthetase levels and histological and mRNA evidence of PNI (Figure 2B,C) [28]. Moreover, the study found that L-asparagine was the only amino acid with high sensitivity and specificity in diagnosing PNI, and patients with high asparagine synthetase exhibited PNI-positive tumors [28]. More studies are necessary to examine the roles of asparagine and asparagine synthetase in PNI and TI for the potential of developing future therapeutics targeting these pathways.

#### 3.2.3. Serine and Glycine

Serine not only promotes cancer-cell proliferation, but also aids in maintaining the health of neurites. Nusser et al. observed that NGF treatment activates protein kinase A, leading to the phosphorylation of RhoA, a G protein that regulates cell-cycle progression, gene expression, and cell motility [213], on serine [214]. This cascade inhibits the RhoA-Rho-associated kinase binding necessary to restrict neurite outgrowth [214]. Moreover, a more recent in vitro analysis found that NGF-induced serine phosphorylation promotes signal transducer and activator of transcription 3 (STAT3), which is then responsible for induced neurite outgrowth [215]. While limited studies exist, Tapia et al. discovered glycine (50 µM) activates chloride-ion membrane currents of neurons in vitro, and glycine receptor activation correlates to depolarizing excitatory potentials [216]. Interestingly, glycine was found to affect neurite outgrowth in a dose-dependent manner, but neurons become desensitized after long exposure, e.g., higher concentrations are needed to induce outgrowth [216].

Banh et al. published the first documented study implicating the role of serine in tumor innervation [207]. In the study, in vitro pancreatic ductal adenocarcinoma cells upregulated serine synthesis following serine/glycine starvation, which promoted cell growth in a dose-dependent manner [207]. Additionally, in this nutrient-poor environment, axons release amino acids, e.g., serine, to support the health of exogenous serine-dependent cancer cells, and in vivo, pancreatic tumors in mice starved of serine/glycine showed increased innervation by sympathetic and sensory nerves [207]. The group found that blocking innervation via NGF receptor TRK inhibitor, LOXO-101, slowed tumor growth in serine/glycine-deprived mice, suggesting the promise of utilizing serine/glycine starvation with innervation inhibitors as cancer therapies [207]. However, more studies should be conducted to prove its therapeutic efficacy in other tumor types and the clinical setting.

### 3.3. Proposed Targets for Cancer–Nerve Crosstalk Research

#### 3.3.1. Lactate

In the exercise sciences, it is understood that muscles release lactate (and its protonated form lactic acid) into the bloodstream following physical activity [217]. Studies have shown that while some lactate is processed by the liver and utilized for oxidative respiration, lactate can also cross the blood–brain barrier and mediate cognitive function [218]. Once imported into neurons by MCT1 in vivo, lactate signals silent information regulator-1 (SIRT1) to induce upregulation of transcriptional factor PGCa and secreted factor FNDC5 to ultimately facilitate BDNF expression [218]. A recent meta-analysis of interval training, characterized by increased lactate output, concluded that physical activity induces BDNF release in humans [219]. While no distinct link between lactate and cancer–nerve crosstalk has been confirmed, BDNF is a known contributor of both PNI [220] and TI [88]; therefore, continued examinations of the role of lactate in this relationship are warranted.

Alternatively, Sandforth et al. found a distinct correlation between MCT1, which imports lactate, and PNI in pancreatic cancer (Figure 2D) [209]. In humans, MCT1 has been found in oligodendrocytes, astrocytes, microglia, endothelial cells, and neurons, and the wellbeing of glia–neuron metabolic crosstalk relies on MCT functionality [221]. Additionally, Lin et al. showed that LDH-A release from damaged neurons facilitates angiogenesis in the central nervous system via interaction with adjacent vimentin-expressing endothelial cells [222]. Further studies should be conducted to fully comprehend the value of lactate, its transporters, and LDH in cancer–nerve crosstalk and investigate the efficacy of lactate silencing/inhibition as a treatment option. 

#### 3.3.2. Vitamin A

Retinoic acid and NGF have been reported to show synergistic effects on neuroprotection related to neuronal survival and growth [223]. Combination treatment of NGF and all-trans retinoic acid (ATRA) on 8705-C thyroid papillary tumor cells inhibited their proliferation and invasion [224]. Arrieta et al. investigated the effect of ATRA on chemotherapy-induced peripheral neuropathy in the male Wistar rat model [225] by administering 20 mg/kg per os (PO) of ATRA for 15 days. ATRA was found to suppress chemotherapy-induced neuropathy by increasing NGF and retinoic-acid-receptor beta (RAR-β) expression. Additionally, the group conducted a randomized, double-blinded, controlled study in which 95 NSCLC patients were administered 20 mg/m^2^ of ATRA per day for 1 week before undergoing chemotherapy over two courses (21 days per course) [225]. ATRA-treated patients presented with a reduction in axonal degeneration [225]. Higashi et al. investigated the effect of retinoic acid on neuroblastoma cell lines and found that treatment with 1 to 10 μM of 13-cis-retinoic acid for 3 to 12 days upregulated cell expression of chromodomain-helicase DNA-binding protein 5 (CHD5), a tumor-suppressing gene, and induced neuronal differentiation in SH-SY5Y, NGP, and SK-N-DZ cells [226]. NGF also increased CHD5 expression and neuronal differentiation in SY5Y cells. The study concluded that 13-cis-retinoic acid administration for a year after surgery has a preventive effect on recurrence in neuroblastoma patients [227]. As NGF is one of the key players in cancer–nerve crosstalk, the link between retinoic acid and NFG in cancer of the nervous system calls for future studies to determine the full extent of this relationship [81,82,83].

#### 3.3.3. Vitamin B

Neurodegenerative disease is caused by alterations in the central nervous system and has been credited to protein misfolding induced by disordered metabolite control of proteins [228]. Protein misfolding is the cause of many types of neurodegenerative diseases, including Alzheimer’s disease (AD) [229]. Studies have presented NGF as a beneficial treatment of AD due its role in promoting neurite outgrowth [230]. Therefore, Ina and Kamei probed the mechanism of vitamin B_12_-mediated neurite outgrowth and found that low concentrations of NGF (10 ng/mL), vitamin B_12_ (6 to 100 µM) promoted neurite outgrowth and the differentiation of PC12 pheochromocytoma cells (a type of neuroendocrine tumor) into neuron-like cells [231]. Upon further investigation, the study found that by using protein kinase inhibitors, vitamin B_12_ stimulates PC12 differentiation in a manner that involves the same signal-transduction pathways activated by NGF. The results suggest that vitamin B_12_ can stimulate neural differentiation, and that like NGF, it stimulates the mitogen-activated protein kinase/extracellular receptor kinase (MAPK/ERK)-signaling pathway [231]. In addition, Okada et al. showed that MeCbl (≥100 nM) promotes neurite outgrowth and neuronal survival [232]. These outcomes were mediated by the methylation cycle and demonstrated that MeCbl increases ERK1/2 and AKT activities through this process [232]. Neurotrophins, such as NGF and BDNF, also activate ERK1/2 and AKT [233]. This activation of ERK1/2 promotes neurite outgrowth and AKT initiates branching of dorsal root ganglia neurites [234]. In a follow-up study, Okada et al. revisited this mechanism and determined that MeCbl increases mTOR activity, a protein kinase that regulates neurite outgrowth and nerve regeneration, through the activation of AKT, which in turn promotes neurite outgrowth in cerebellar granule neurons [235,236]. 

In addition to vitamin B_12_, other isoforms of vitamin B are linked to promoting peripheral nerve regeneration, including B_1_ and B_6_ [208]. These neurotrophic B vitamins support the development of new cell structures and are key players in nerve regeneration while also maintaining neuronal viability. Vitamin B_1_ acts as a site-directed antioxidant and facilitates the use of carbohydrates for energy production, vitamin B_6_ balances nerve metabolism, and vitamin B_12_ promotes neural cell survival and myelin sheath maintenance [236]. It has been determined that a combination of vitamin B complex is necessary to optimize regeneration in cases of peripheral neuropathy [237]. Deficiencies in these vitamins have been associated with nerve dysfunction and damage and can lead to peripheral neuropathy [238]. Altun and Kurutaş suggested that tissue vitamin B complex levels vary during crush-induced peripheral nerve injury, and supplementation during these acute time periods may accelerate nerve regeneration [239]. These neuroregulatory properties of vitamin B, especially vitamin B_12_ (in an NGF-like manner) [231] show the potential value of targeting vitamin B in PNI and TI therapies. However, further research may provide a clearer mechanism by which vitamin B contributes to this crosstalk.

#### 3.3.4. Vitamin D

The active form of vitamin D, 1,25(OH)_2_D_3_, is a hormone that has a similar influence as that of neurosteroids [240]. Vitamin D has an important role in neuronal differentiation and maturation via neurotrophin regulation [241]. NGF, glial-cell-derived neurotrophic factor (GDNF), and neurotrophin 3 levels are increased by vitamin D, to facilitate neuronal growth [242]. Male Sprague Dawley rats injected with vitamin D_3_ for 8 days showed significant increased GDNF in the cortex with significantly decreased infarction amount after middle cerebral-artery ligation [120]. Rats from mothers with vitamin D_3_ deficiency showed significant changes in brain structure and low NGF and GDNF levels at birth [243]. Even though there has been no apparent link between vitamin D and cancer–nerve crosstalk, the value of vitamin D in controlling neurotrophin expression [241] highlights its potential in mediating PNI and/or TI and requires further investigation.

#### 3.3.5. Vitamin E and K

Vitamin E has an important role in maintaining normal neurological function and structure [244]. In in vivo sciatic-nerve-crush injury models, vitamin E acetate showed a neuroprotective effect with steady improvement in motor-nerve-conduction velocity and thermal hyperalgesia [245]. In addition, Pace et al. showed that vitamin E significantly reduced the incidence of chemotherapy-induced peripheral neuropathy and its severity in patients following cisplatin chemotherapy, without affecting the antitumorigenic activity of cisplatin [246]. These neuroprotective properties of vitamin E suggest its potential involvement in cancer–nerve crosstalk, and therefore warrant additional studies. Vitamin K_1_ and K_2_ treatment of PC12D cells (100 µg/mL) in the presence of 2.5 to 50 ng/mL NGF significantly increased neurite outgrowth [247]. One study found that vitamin K-induced neurite outgrowth is potentially mediated via protein kinase A and MAPK cascades [248]. In the nervous system, vitamin K-dependent growth-arrest-specific protein 6 activates receptors of TAM, which includes tyrosine-protein kinase receptor 3, receptor tyrosine kinase, and MER proto-oncogene tyrosine kinase [249]. Similar defects in TAM are associated with cancer, indicating the potential for vitamin K to facilitate cancer initiation [249]. More studies are needed to fully understand the mechanisms and contributions of vitamin K in the nervous system, cancer progression, and cancer–nerve crosstalk.

## 4. Conclusions

In this review, we have summarized the current knowledge of significant yet relatively underappreciated metabolites in cancer development and metastasis (i.e., lactate, vitamins A, B, C, D, E and K, asparagine, arginine, serine, glycine, and cysteine). In addition, we discussed metabolites and their regulators currently established as contributors to cancer–nerve crosstalk (i.e., SLC2A3, asparagine synthetase, and serine) and suggested metabolites that may be implicated in cancer–nerve crosstalk based on their tumorigenic and neuroregulatory properties (i.e., lactate and vitamins A, B, D, E, and K). However, there remains limited information to make a clear connection between these nutrients and cancer–nerve crosstalk. It is our hope that this review serves researchers as a guide to developing future studies to determine the roles of metabolites in cancer and nerve crosstalk in order to expand the collective understanding of cancer mechanisms that may be beneficial in developing potential therapies for cancer, PNI, and TI.

## Figures and Tables

**Figure 1 nutrients-14-01722-f001:**
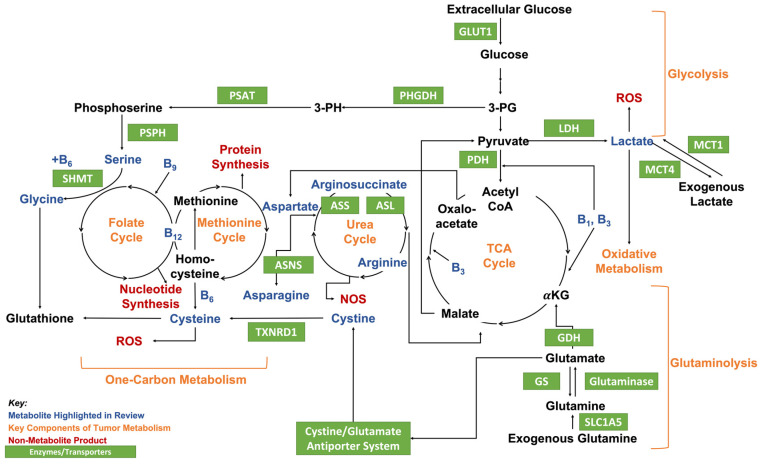
Complex mechanisms induced by glucose, glutamine, and lactate uptake encompass the metabolism of tumor cells. 3-PG: 3-phosphoglycerate, 3-PH: 3-phosphohydroxypyruvate, ASL: arginosuccinate lyase, ASNS: asparagine synthetase, ASS: arginosuccinate synthase, αKG: alpha-ketoglutarate, GDH: glutamine dehydrogenase, GS: glutamine synthetase, LDH: lactate dehydrogenase, MCT1/4: monocarboxylate transporter 1/4, NOS: nitric oxide synthase, PDH: pyruvate dehydrogenase, PHGDH: phosphoglycerate dehydrogenase, PSAT: phosphoserine aminotransferase, PSPH: phosphoserine phosphatase, ROS: reactive oxygen species, SHMT: serine hydroxymethyltransferase, TXNRD1: thioredoxin reductase 1.

**Figure 2 nutrients-14-01722-f002:**
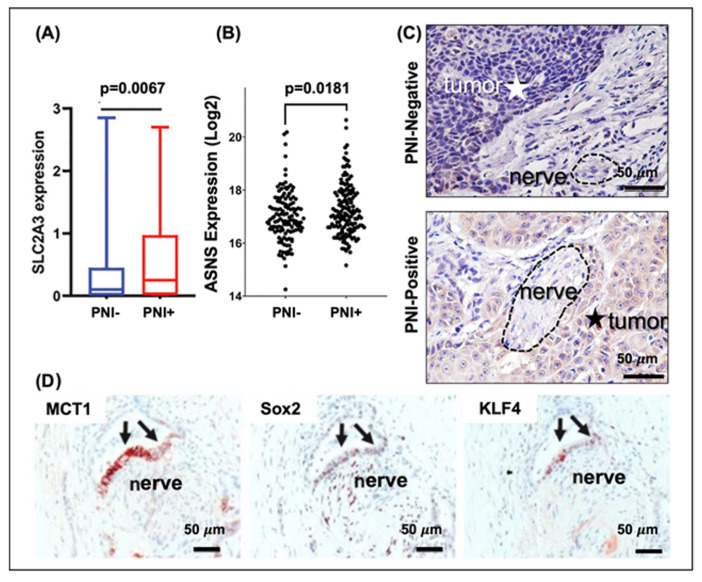
Metabolism-related cancer–nerve crosstalk. (**A**) Vitamin C transporter SLC2A3 is shown via immunohistochemistry to be upregulated in colorectal cancer patients with perineural invasion (PNI) [206]. mRNA (**B**) and immunohistochemistry (**C**) analyses found that asparagine synthetase (ASNS) is upregulated in PNI-positive oral squamous-cell carcinoma patients. Dotted circles represent nerve trunks, and stars indicate the tumor region [28]. (**D**) Lactate importer MCT1 is colocalized with Sox2- and KLF-positive (cell-proliferation markers) in cases of PNI in pancreatic adenocarcinoma [205]. Scale is 50 μm. Figures are modified from Gao et al., Fu et al., and Sandforth et al., respectively. Figure rights for reuse are available via the Creative Commons Attribution (CC BY) License.

**Table 1 nutrients-14-01722-t001:** Metabolite contribution to tumor survival varies between cancer types.

Metabolite	Cancer Type	Role in Tumor Progression
Vitamin A	Breast	4-HPR induces cell death [37]; vitamin A and retinol reduce risk [105]
	Colon/Colorectal	4-HPR induces cell death [37]
	Head/Neck	4-HPR induces cell death [37]
	Gastric	Inhibits polycyclic hydrocarbon-induced carcinomas [37,108]
	Lung	Blood levels of α- and β-carotene, total carotenoids, and retinol are inversely associated with cancer risk [109]
	Prostate	4-HPR induces cell death [37]
Vitamin B_1_	Breast	Intermediate concentrations promote Ehrlich’s ascites proliferation in thiamine-deficient patients; high concentrations inhibit proliferation [110]; patients exhibit decreased expression of SLC9A3 transporter gene [111,112]
	Head/Neck	Patients exhibit decreased expression of SLC9A3 transporter gene [112] Intake reduces risk of esophageal cancer [103]
Vitamin B_3_	Head/Neck	Intake reduces risk of esophageal cancer [103]
Vitamin B_6_	Head/Neck	Intake reduces risk of esophageal cancer [103]
Vitamin B_9_	Head/Neck	Intake reduces risk of esophageal cancer [103]
Vitamin B_12_	Head/Neck	Intake increases risk of esophageal cancer [103]
	Leukemia/Lymphoma	Elevated plasma levels associated with 1-year cancer risk [113]
	Liver	Elevated plasma levels associated with 1-year cancer risk [113]
	Lung	Positively associated with cancer risk in dose-dependent manner [114]
	Pancreatic	Elevated plasma levels associated with 1-year cancer risk [113]
Vitamin C	Breast	Low concentrations induce cell invasiveness; high doses restrict EMT [115]
	Skin	Low doses reduce cell viability and invasiveness; high doses promote cell migration [116]
Vitamin D	Breast	Calcitroil and D3 analogs suppress MMP-2 and -9 and VCAM-1; low serum D3 levels are associated with high incidence [117]
	Colon/Colorectal	Low serum D3 levels are associated with high incidence [117]; analog PRI-2191 enhances ability of 5-FU to restrict cell cycle [118]; serum levels of ≥33 ng/mL correlates with a 50% decreased risk [119]
	Gastric	Low serum D3 levels associated with high incidence [117]
	Head/Neck	MART-10 induces cell-cycle arrest and suppresses p21 and p27 [120]
	Prostate	Lower serum levels are associated with an increased risk; D3 and analogs inhibit invasiveness and expression of MMP-2 and -9 and VCAM-1 [117]
Vitamin E	Breast	Tocotrienols exhibit chemotherapeutic and antitumor properties [41]; γ-tocotrienol induces cell-cycle arrest [121,122]
	Colon/Colorectal	Tocotrienols exhibit antitumor properties [41]; γ-tocotrienol mediates apoptosis via apoptotic caspase-3 and NFκB suppression [123]
	Liver	Tocotrienols exhibit chemotherapeutic properties [41]
	Lung	Tocotrienols exhibit chemotherapeutic properties [41]
	Pancreatic	Tocotrienols exhibit chemotherapeutic properties [41]
	Prostate	Tocotrienols exhibit chemotherapeutic properties [41]
Vitamin K	Breast	K_2_ induces nonapoptotic cell death [42]
Arginine	Breast	Low plasma levels act as a prognostic biomarker [124]; arginine starvation is used to treat arginosuccinate synthase-deficient patients [101,124]
	Ovarian	Cancer cells are deficient in arginosuccinate synthase-1; ADI-PED-20 is used to degrade arginine [125]
	Prostate	Low plasma levels act as a prognostic biomarker [126]
	Skin	Cells are deficient in arginosuccinate synthase-1; ADI-PEG20-resistant cancer exhibits c-MYC binding to the promoter of arginosuccinate synthase-1 [26]
Asparagine	Breast	Maintains health of glutamine-independent cells [96]
	Cervical	Facilitates mTOR activation in the absence of glutamine [96]
	Liposarcoma	Maintains health of glutamine-independent cells [96]
Cysteine	Breast	Inhibition of histone deacetylase-6 sensitizes TNBC cells to cysteine deprivation via cystine/glutamate antiporter-targeted therapies [98]
	Colon/Colorectal	Starvation induces a reduction in liver-metastatic cell proliferation [29]
	Pancreatic	The deletion of cystine transporter gene SLC7A11 inhibits autophagy and diminishes cysteine homeostasis [127]
Glycine	Colon/Colorectal	Metabolism increases when starved of serine [128]; serine–glycine inhibition should be used in conjunction to decrease tumor size [102]
	Lung	De novo serine and glycine are allocated to glutathione synthesis [128]
Lactate	Breast	10 mM L-lactate acts as chemoattractant and facilitates migration [129]; intermediate and high supplementation upregulates oncogene, proliferation gene, tumor suppressor, and transcription-factor expression [130];
	Cervical	When given glucose and lactate, oxidative cancer cells prefer lactate; cells thrive when given lactate supplementation; oxidative cells exhibit high expression of MCT1 versus MCT4; MCT1 inhibition induces necrosis in oxidative cells [25]; DLAD targets metabolism [131]
	Colon/Colorectal	Glycolytic cells fail to thrive upon glucose starvation with lactate supplementation [25]
	Head/Neck	DLAD targets metabolism [131]
	Lung	NFκB signaling [132]; LDH-A inhibition sensitizes cells to radiation [133]
	Skin	DLAD targets metabolism [131]
Serine	Breast	Cells prefer serine over glycine and exhibit a decrease in nucleic acid synthesis when starved of serine [99]; brain metastatic cells upregulate de novo serine when starved of exogenous alternative [134]
	Colon/Colorectal	Cells prefer serine over glycine and exhibit a decrease in nucleic acid synthesis when starved of serine [99]; metabolism increases when starved of serine; starvation decreases YAP activation [135]; serine–glycine catabolism induced by stress promotes formate production [136]; serine–glycine inhibition should be used in conjunction to decrease tumor size; starvation induces metabolic stress and p53-activated glycolysis [102]
	Lung	Promotes purine synthesis in cancer cells; de novo serine and glycine are allocated to glutathione synthesis [128]

## Data Availability

Not applicable.

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
