# Peer review of "Exploring the Role of Metabolites in Cancer and the Associated Nerve Crosstalk"

_nutrients, 2022, doi:10.3390/nu14091722_

Round 1

Reviewer 1 Report

The article entitled “Exploring the Role of Metabolites in Cancer and the Associated Nerve Crosstalk”  the authors discuss relatively underappreciated metabolites in cancer-nerve crosstalk, including lactate, vitamins, and amino acids, and propose the investigation of nutrients in cancer-nerve crosstalk based on their tumorigenicity and neuroregulatory capabilities. It is a review study.

Even though it is a review article, I make some suggestions:

- In the Abstract: Even if it is a review and discussion, the abstract can be rewritten including the clinical importance of the same along with the research objective

- What inclusion and exclusion criteria were used?

- Databases?

- As it is a review, through the inclusion and exclusion criteria, I suggest a reduction and alteration of the data presented in table 1. Perhaps it would be interesting to insert a column with the main result of each analyzed article and not only the reference.

- Given the information collected in the literature, what is the group's critical opinion in relation to the results?

- What would be the contribution to the community and the impact?

- What is the potential of clinical applicability? Maybe add the conclusion.

Reviewer 2 Report

In this review manuscript, Gu et al. summarized the current understanding of the roles of metabolites in cancer and the cancer-nerve crosstalk. This is a very interesting topic and the manuscript was very well written. However, this manuscript can be significantly improved if the authors can provide some updates, supposing the information is available in the literature.

[1]  The nerve-cancer crosstalk is very interesting. However, this manuscript did not properly address the real "crosstalk". I suggest the author's update information about the role of nerves in the tumor microenvironment and their impact on cancer progression and treatment.

[2]  some sections can be simplified to retain the interest of readers in finishing reading the whole manuscript. For example, the information and numbers in the brackets in the paragraph (Lines 169-187) can be reorganized in a table. Line 265, "12.5, 25, 37.5, 50, and 75 times " can be simplified from 12.5 times to 75 times.

[3] The title of the manuscript should be slightly modified: "role" to "roles"; "the associated nerve crosstalk" to "nerve-cancer cell crosstalk".

[4] The manuscript was very well written. However, small grammar errors need to be fixed. For example, 10mM should be 10 mM (Line 140). Please also check Lines 441,443,444, 353-354.
